# Domain Adaptive Multibranch Networks

**Róger Bermúdez-Chacón, Mathieu Salzmann, & Pascal Fua**
Computer Vision Laboratory
École Polytechnique Fédérale de Lausanne
Station 14, CH-1015 Lausanne, Switzerland
`{roger.bermudez,mathieu.salzmann,pascal.fua}@epfl.ch`

## Abstract

We tackle unsupervised domain adaptation by accounting for the fact that different domains may need to be processed differently to arrive to a common feature representation effective for recognition. To this end, we introduce a deep learning framework where each domain undergoes a different sequence of operations, allowing some, possibly more complex, domains to go through more computations than others. This contrasts with state-of-the-art domain adaptation techniques that force all domains to be processed with the same series of operations, even when using multi-stream architectures whose parameters are not shared. As evidenced by our experiments, the greater flexibility of our method translates to higher accuracy. Furthermore, it allows us to handle any number of domains simultaneously.

## 1 Introduction

While deep learning has ushered in great advances in automated image understanding, it still suffers from the same weaknesses as all other machine learning techniques: when trained with images obtained under specific conditions, deep networks typically perform poorly on images acquired under different ones. This is known as the domain shift problem: the changing conditions cause the statistical properties of the test, or target, data, to be different from those of the training, or source, data, and the network's performance degrades accordingly.

Domain adaptation aims to address this problem, especially when annotating images from the target domain is difficult, expensive, or downright infeasible. The dominant trend is to map images to features that are immune to the domain shift, so that the classifier works equally well on the source and target domains (Fernando et al., 2013; Ganin & Lempitsky, 2015; Sun & Saenko, 2016). In the context of deep learning, the standard approach is to find those features using a *single architecture* for both domains (Tzeng et al., 2014; Ganin & Lempitsky, 2015; Sun & Saenko, 2016; Yan et al., 2017; Zhang et al., 2018). Intuitively, however, as the domains have different properties, it is not easy to find one network that does this effectively for both. A better approach is to allow domains to undergo different transformations to arrive at domain-invariant features. This has been the focus of recent work (Tzeng et al., 2017; Bermúdez-Chacón et al., 2018; Rozantsev et al., 2018; 2019), where source and target data pass through *two different networks* with the same architecture but different weights, nonetheless related to each other.

In this paper, we introduce a novel, even more flexible paradigm for domain adaptation, that allows the different domains to undergo *different computations*, not only in terms of layer weights but also in terms of number of operations, while *selectively sharing* subsets of these computations. This enables the network to automatically adapt to situations where, for example, one domain depicts simpler images, such as synthetic ones, which may not need as much processing power as those coming from more complex domains, such as images taken in-the-wild. Our formulation reflects the intuition that source and target domain networks should be similar because they solve closely related problems, but should also perform domain-specific computations to offset the domain shift.

To turn this intuition into a working algorithm, we develop a multibranch architecture that sends the data through multiple network branches in parallel. What gives it the necessary flexibility are trainable gates that are tuned to modulate and combine the outputs of these branches, as shown in Fig. 1. Assigning to each domain its own set of gates allows the global network to learn what set of

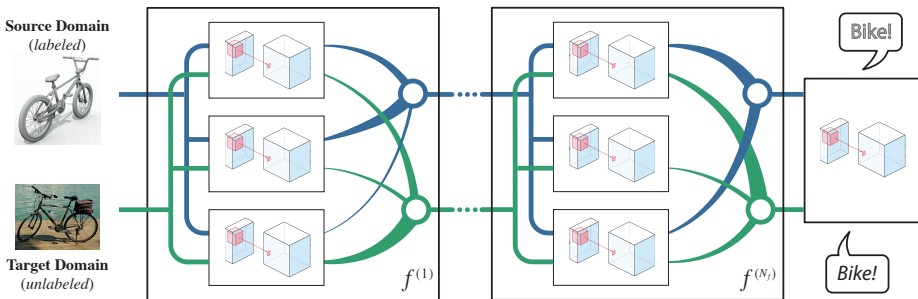

Figure 1: A **Domain Adaptive Multibranch Network** is a sequence of computational units $f^{(i)}$, each of which processes the data in parallel branches, whose outputs are then aggregated in a weighted manner by a gate to obtain a single response. To allow for domain-adaptive computations, each domain has its own set of gates, one for each computational unit, which combine the branches in different ways. As a result, some computations are shared across domains while others are domain-specific.

computations should be carried out for each one. As an additional benefit, in contrast to previous strategies for untying the source and target streams (Rozantsev et al., 2018; 2019), our formulation naturally extends to more than two domains.

In other words, our contribution is a learning strategy that adaptively adjusts the specific computation to be performed for each domain. To demonstrate that it constitutes an effective approach to extracting domain-invariant features, we implement it in conjunction with the popular domain classifier-based method of Ganin & Lempitsky (2015). Our experiments demonstrate that our Domain Adaptive Multibranch Networks, which we will refer to as DAMNets, not only outperform the original technique of Ganin & Lempitsky (2015), but also the state-of-the-art strategy for untying the source and target weights of Rozantsev et al. (2019), which relies on the same domain classifier. We will make our code publicly available upon acceptance of the paper.

## 2 RELATED WORK

**Domain Adaptation.** Domain adaptation has achieved important milestones in recent years (Dai et al., 2007; Gretton et al., 2009; Pan et al., 2010; Fernando et al., 2013; Sun et al., 2016; Shu et al., 2018), with deep learning-based methods largely taking the lead in performance. The dominant approach to deep domain adaptation is to learn a domain-invariant data representation. This is commonly achieved by finding a mapping to a feature space where the source and target features have the same distribution. In Tzeng et al. (2014); Long et al. (2015; 2017); Yan et al. (2017), the distribution similarity was measured in terms of Maximum Mean Discrepancy (Gretton et al., 2007), while other metrics based on second- and higher-order statistics were introduced in Sun & Saenko (2016); Koniusz et al. (2017); Sun et al. (2017). In Saito et al. (2018), the distribution alignment process was disambiguated by exploiting the class labels, and in Häusser et al. (2017); Shkodrani et al. (2018) by leveraging anchor points associating embeddings between the domains. Another popular approach to learning domain-invariant features is to train a classifier to recognize the domain from which a sample was drawn, and use adversarial training to arrive to features that the classifier can no longer discriminate (Tzeng et al., 2015; Ganin et al., 2016; 2017). This idea has spawned several recent adversarial domain adaptation classification (Hu et al., 2018; Zhang et al., 2018), semantic segmentation (Hoffman et al., 2018; Chen et al., 2018; Hong et al., 2018), and active learning (Su et al., 2019) techniques, and we will use such a classifier.

Closest in spirit to our approach are those that do not share the weights of the networks that process the source and target data (Tzeng et al., 2017; Bermúdez-Chacón et al., 2018; Rozantsev et al., 2018; 2019). In Tzeng et al. (2017), the weights were simply allowed to vary freely. In Rozantsev et al. (2018); Bermúdez-Chacón et al. (2018), it was shown that regularizing them to remain close to each other was beneficial. More recently, Rozantsev et al. (2019) proposed to train small networks to map the source weights to the target ones. While these methods indeed untie the source and target weights, the source and target data still undergo the same computations, i.e., number of operations.

In this paper, we argue that the amount of computation, that is, the network capacity, should adapt to each domain and reflect their respective complexities. We rely on a domain classifier as in Tzeng

et al. (2015); Ganin et al. (2016; 2017). However, we do not force the source and target samples to go through the same transformations, which is counterintuitive since they display different appearance statistics. Instead, we start from the premise that they should undergo different computations and use domain-specific gates to turn this premise into our DAMNet architecture.

**Dynamic Network Architectures.** As the performance of a neural network is tightly linked to its structure, there has been a recent push towards automatically determining the best architecture for the problem at hand. While neural architecture search techniques (Zoph & Le, 2017; Liu et al., 2018; 2019; Pham et al., 2018; Zoph et al., 2018; Real et al., 2019; Noy et al., 2019) aim to find one fixed architecture for a given dataset, other works have focused on dynamically adapting the network structure at inference time (Graves, 2016; Ahmed & Torresani, 2017; Shazeer et al., 2017; Veit & Belongie, 2018; Wu et al., 2018). In particular, in Ahmed & Torresani (2017); Shazeer et al. (2017); Veit & Belongie (2018); Bhatia et al. (2019), gates were introduced for this purpose. While our DAMNets also rely on gates, their role is very different: first, we work with data coming from different domains, whereas these gated methods, with the exception of Bhatia et al. (2019), were all designed to work in the single-domain scenario. Second, and more importantly, these techniques aim to define a different computational path for every test *sample*. By contrast, we seek to determine the right computation for each *domain*. Another consideration is that we freeze our gates for inference while these methods must constantly update theirs. We believe this to be ill-suited to domain adaptation, particularly because learning to adapt the gates for the target domain, for which only unlabeled data is available, is severely under-constrained. This lack of supervision may be manageable when one seeks to define operations for a whole domain, but not when these operations are sample-specific.

## 3 METHOD

We now describe our deep domain adaptation approach, which automatically adjusts the computations that the different domains undergo. We first introduce the multibranch networks that form the backbone of our DAMNet architecture and then discuss training in the domain adaptation scenario.

### 3.1 MULTIBRANCH NETWORKS

Let us first consider a single domain. In this context, a traditional deep neural network can be thought of as a sequence of $N_f$ operations $f^{(i)}(\cdot)_{1 \leq i \leq N_f}$, each transforming the output of the previous one. Given an input image $\mathbf{x}$, this can be expressed as

$$\mathbf{x}^{(0)} = \mathbf{x}$$
$$\mathbf{x}^{(i)} = f^{(i)}(\mathbf{x}^{(i-1)}). \quad (1)$$

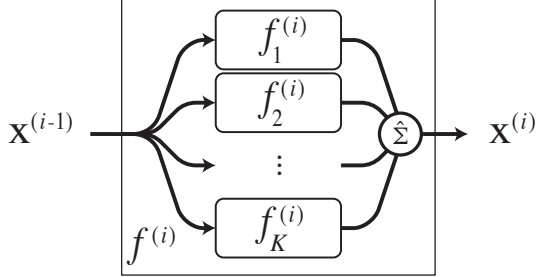

Figure 2: A **computational unit** $f^{(i)}$ is an aggregation of the outputs of parallel computations, or branches, $f_j^{(i)}$.

As a general convention, each operation $f^{(i)}(\cdot)$ can represent either a single layer or multiple ones. Our formulation extends this definition by replacing each $f^{(i)}$ by multiple parallel computations, as shown in Fig. 2. More specifically, we replace each $f^{(i)}$ by a *computational unit* $\{f_1^{(i)}, \ldots, f_K^{(i)}\}$ consisting of $K$ parallel *branches*. Note that this $K$ can be different at each stage of the network and should therefore be denoted as $K^{(i)}$. However, to simplify notation, we drop this index below. Given this definition, we write the output of each computational unit as

$$\mathbf{x}^{(i)} = \hat{\Sigma} \left( f_1^{(i)}(\mathbf{x}^{(i-1)}), \ldots, f_K^{(i)}(\mathbf{x}^{(i-1)}) \right), \quad (2)$$

where $\hat{\Sigma}(\cdot)$ is an aggregation operator that could be defined in many ways. It could be a simple summation that gives all outputs equal importance, or, at the opposite end of the spectrum, a multiplexer that selects a single branch and ignores the rest. To cover the range between these two alternatives,

we introduce *learnable gates* that enable the network to determine what relative importance the different branches should be given. Our gates perform a weighted combination of the branch outputs. Each gate is controlled by a set of $K$ *activation weights* $\{\phi_j^{(i)}\}_{1 \leq j \leq K}$, and a unit returns

$$\mathbf{x}^{(i)} = \sum_{j=1}^{K} \phi_j^{(i)} \cdot f_j^{(i)}(\mathbf{x}^{(i-1)}) . \tag{3}$$

If $\forall j, \phi_j^{(i)} = 1$, the gate performs a simple summation. If $\phi_j^{(i)} = 1$ for a single $j$ and 0 for the others, it behaves as a multiplexer. The activation weights $\phi_j^{(i)}$ enable us to modulate the computational graph of network block $f^{(i)}$. To bound them and encourage the network to either select or discard each branch in a computational unit, we write them in terms of sigmoid functions with adaptive steepness. That is,

$$\phi_j^{(i)} = \left(1 + \exp\left(-\pi^{(i)} \cdot g_j^{(i)}\right)\right)^{-1} , \tag{4}$$

where the $g_j^{(i)}$s are learnable unbounded model parameters, and $\pi^{(i)}$ controls the *plasticity* of the activation—the rate at which $\phi_j^{(i)}$ varies between the extreme values 0 and 1 for block $i$. During training, we initially set $\pi^{(i)}$ to a small value, which enables the network to explore different gate configurations. We then apply a cooling schedule on our activations, by progressively increasing $\pi^{(i)}$ over time, so as to encourage the gates to reach a firm decision. Note that our formulation does not require $\sum_{j=1}^{K} \phi_j^{(i)} = 1$, that is, we do not require the aggregated output $\mathbf{x}^{(i)}$ to be a convex combination of the branch outputs $f_j^{(i)}(\mathbf{x}^{(i-1)})$. This is deliberate because allowing the activation weights to be independent from one another provides additional flexibility for the network to learn general additive relationships.

Finally, a *Multibranch Network* is the concatenation of multiple computational units, as shown in Fig. 1. For the aggregation within each unit $f^{(i)}$ to be possible, the $f_j^{(i)}$s' outputs must be of matching shapes. Furthermore, as in standard networks, two computational units can be attached only if the output shape of the first one matches the input shape of the second. Although it would be possible to define computational units at any point in the network architecture, in practice, we usually take them to correspond to groups of layers that are semantically related. For example, one would group a succession of convolutions, pooling and non-linear operations into the same computational unit.

## 3.2 DOMAIN ADAPTIVE MULTIBRANCH NETWORKS

### 3.2.1 TWO DOMAINS

Our goal is to perform domain adaptation, that is, leverage a large amount of labeled images, $\mathbf{X}^s = \{\mathbf{x}_1^s, \ldots, \mathbf{x}_N^s\}$ with corresponding annotations $\mathbf{Y}^s = \{\mathbf{y}_1^s, \ldots, \mathbf{y}_N^s\}$, drawn from a source domain, to train a model for a target domain, whose data distribution is different and for which we only have access to unlabeled images $\mathbf{X}^t = \{\mathbf{x}_1^t, \ldots, \mathbf{x}_M^t\}$.

To this end, we extend the gated networks of Section 3.1 by defining two sets of gates, one for the source domain and one for the target one. Let $\{(\phi^s)_j^{(i)}\}_{j=1}^{K}$ and $\{(\phi^t)_j^{(i)}\}_{j=1}^{K}$ be the corresponding source and target activation weights for computational unit $f^{(i)}$, respectively. Given a sample $\mathbf{x}^d$ coming from a domain $d \in \{s, t\}$, we take the corresponding output of the $i$-th computational unit to be

$$(\mathbf{x}^d)^{(i)} = \sum_{j=1}^{K} (\phi^d)_j^{(i)} \cdot f_j^{(i)}\left((\mathbf{x}^d)^{(i-1)}\right) . \tag{5}$$

Note that under this formulation, the domain identity $d$ of the sample is required in order to select the appropriate $(\phi^d)^{(i)}$.

The concatenated computational units forming the DAMNet encode sample $\mathbf{x}$ from domain $d$ into a feature vector $\mathbf{z} = f(\mathbf{x}, d)$. Since the gates for different domains are set independently from

one another, the outputs of the branches for each computational unit are combined in a domain-specific manner, dictated by the activation weights $(\phi^d)_j^{(i)}$. Therefore, the samples are encoded to a common space, but arrive to it through potentially different computations. Fig. 3 depicts this process. Ultimately, the network can learn to share weights for computational unit $f^{(i)}$ by setting $(\phi^s)_j^{(i)} = (\phi^t)_j^{(i)}, \forall j$. It can also learn to fully untie the weights by having $A_i^S \cap A_i^T = \emptyset$, where $A_i^S$ and $A_i^T$ denote the set of non-zero activations in the two domains. Finally, in contrast to Tzeng et al. (2017); Bermúdez-Chacón et al. (2018); Rozantsev et al. (2018; 2019), it can learn to use more computation for one domain than for the other by setting $(\phi^s)_j^{(i)} > 0$ for two different branches $f_j^{(i)}$ while having only a single non-zero $(\phi^t)_j^{(i)}$, for a particular computational unit $f^{(i)}$.

The above formulation treats all branches for each computational unit as potentially sharable between domains. However, it is sometimes desirable not to share *at all*. For example, *batch-normalization layers* that accumulate and update statistics of the data over time, even during the forward pass, are best exposed to a single domain to learn domain-specific statistics. We allow for this by introducing computational units whose gates are fixed, yet domain specific, and that therefore act as multiplexers.

After the last computational unit, a small network $p_y$ operates directly on the encodings and returns the class assignment $\hat{\mathbf{y}} = p_y(\mathbf{z})$, thus subjecting the encodings for all samples to the same set of operations.

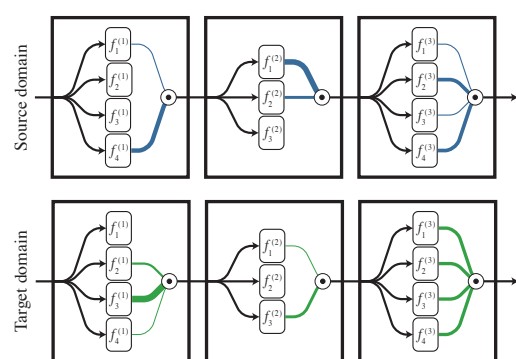

Figure 3: **Computational graphs** for the source (top) and target (bottom) domains, *for the same network.* While both domains share the same computational units, their outputs are obtained by different aggregations of their inner operations, e.g., in the first unit, the source domain does not use the middle two operations, whereas the target domain does; by contrast, both exploit the fourth operation. In essence, this scheme adapts the amount of computation that each domain is subjected to.

### 3.2.2 MULTIPLE DOMAINS

The formulation outlined above extends naturally to more than two domains, by assigning one set of gates per domain. This enables us to exploit annotated data from different source domains, and even to potentially handle multiple target domains simultaneously. In this generalized case, we introduce governing sets of gates with activations $\phi^{d_1}, \dots, \phi^{d_D}$ for $D$ different domains. They act in the same way as in the two-domain case and the overall architecture remains similar.

### 3.2.3 TRAINING

When training our models, we jointly optimize the gate parameters $(g^d)_j^{(i)}$, from Eq. 4, along with the other network parameters using standard back-propagation. To this end, we make use of a composite loss function, designed to encourage correct classification for labeled samples from the source domain(s) and align the distributions of all domains, using labeled and unlabeled samples. This loss can be expressed as

$$L_{\text{DAMNet}} = \frac{1}{|\ell|} \sum_{n=1}^{|\ell|} L_y(\mathbf{y}_n, \hat{\mathbf{y}}_n) + \frac{1}{|\ell \cup u|} \sum_{n=1}^{|\ell \cup u|} L_d(\mathbf{d}_n, \hat{\mathbf{d}}_n) , \qquad (6)$$

where $\ell$ and $u$ are the sets of labeled and unlabeled samples, respectively, and where we assumed, without loss of generality, that the samples are ordered.

The first term in this loss, $L_y(\mathbf{y}, \hat{\mathbf{y}})$, is the standard cross-entropy, which compares the ground-truth class probabilities $\mathbf{y}$ with the predicted ones $\hat{\mathbf{y}} = p_y(\mathbf{z})$, where, as discussed in Section 3.2.1, $\mathbf{z} = f(\mathbf{x}, d)$ is the feature encoding of sample $\mathbf{x}$ from domain $d$. For the second term, which encodes distribution alignment, we rely on the domain confusion strategy of Ganin & Lempitsky (2015), which is commonly used in existing frameworks. Specifically, for $D$ domains, we make use of an auxiliary domain classifier network $p_d$ that predicts a $D$-dimensional vector of domain probabilities $\hat{\mathbf{d}}$ given the feature vector $\mathbf{z}$. Following the gradient reversal technique of Ganin & Lempitsky

(2015), we express the second term in our loss as $L_d(\mathbf{d}, \hat{\mathbf{d}}) = -\sum_{i=1}^{D} \mathbf{d}_i \log(\hat{\mathbf{d}}_i)$ , where $\mathbf{d}$ is the $D$-dimensional binary vector encoding the ground-truth domain, $\mathbf{d}_i$ indicates the $i$-th element of $\mathbf{d}$, and $\hat{\mathbf{d}} = p_d(R(\mathbf{z}))$, with $R$ the gradient reversal pseudofunction of Ganin & Lempitsky (2015) that enables to incorporate adversarial training directly into back-propagation. That is, with this loss, standard back-propagation trains jointly the domain classifier to discriminate the domains and the feature extractor $f(\cdot)$ to produce features that fool this classifier.

When training is complete and the gates have reached a stable state, the branches whose activations are close to zero are deactivated. This prevents the network from performing computations that are irrelevant and allows us to obtain a more compact network to process the target data.

## 4    EVALUATION

### 4.1    BASELINES

Since we rely on the domain confusion loss to train our model, we treat the Domain-Adversarial Neural Network (DANN) method of Ganin & Lempitsky (2015), as our first baseline.

To demonstrate the benefits of our approach over simply untying the source and target stream parameters, we compare our approach against the Residual Parameter Transfer (RPT) method of Rozantsev et al. (2019), which constitutes the state of the art in doing so. Note that RPT also relies on the domain confusion loss, which makes our comparison fair. In addition, we report the results of directly applying a model trained on the source domain to the target, without any domain adaptation, which we refer to as "No DA". We also provide the oracle accuracy of a model trained on the fully-labeled target domain, referred to as "*On TD*".

### 4.2    IMPLEMENTATION DETAILS

We adapt different network architectures to the multibranch paradigm for different adaptation problems. For all cases, we initialize our networks' parameters by training the original versions of those architectures on the source domains, either from scratch, for simple architectures, or by fine-tuning weights learned on ImageNet, for very deep ones. We then set the parameters of all branches to the values from the corresponding layers. We perform this training on the predefined training splits, when available, or on 75% of the images, otherwise. The initial values of the gate parameters are defined so as to set the activations to $\frac{1}{K}$, for each of the $K$ branches. This prevents our networks from initially favoring a particular branch for any domain.

To train our networks, we use Stochastic Gradient Descent with a momentum of 0.9 and a variable learning rate defined by the annealing schedule of Ganin & Lempitsky (2015) as $\mu_p = \frac{\mu_0}{(1+\alpha \cdot p)^\beta}$, where $p$ is the training progress, relative to the total number of training epochs, $\mu_0$ is the initial learning rate, which we take to be $10^{-2}$, and $\alpha = 10$ and $\beta = 0.75$ as in Ganin & Lempitsky (2015). We eliminate exploding gradients by $\ell_2$-norm clipping. Furthermore, we modulate the plasticity of the activations at every gate as $\pi^{(i)} = 1 - p$, that is, we make $\pi^{(i)}$ decay linearly as training progresses. As data preprocessing, we apply mean subtraction, as in Ganin & Lempitsky (2015). We train for 200 epochs, during which the network is exposed to all the image data from the source and target domains, but only to the annotations from the source domain(s).

Our "*On TD*" oracle is trained on either the preset training splits, when available, or our defined training data, and evaluated on the corresponding test data. For the comparison to this oracle to be meaningful, we follow the same strategy for our DAMNets. That is, we use the unlabeled target data from the training splits only and report results on the testing splits. This protocol differs from that of Rozantsev et al. (2019), which relied on a transductive evaluation, where *all* the target images, training and test ones, were seen by the networks during training.

### 4.3    IMAGE RECOGNITION

We evaluate our method in the task of image recognition for which we use several domain adaptation benchmark problems: **Digits**, which comprises three domains: MNIST (LeCun et al., 1998), MNIST-M (Ganin & Lempitsky, 2015), and SVHN (Netzer et al., 2011); **Office** (Saenko et al.,

Table 1: **Domain Adaptation datasets and results.** We compare the accuracy of our DAMNet approach with that of DANN (Ganin & Lempitsky, 2015) and of RPT (Rozantsev et al., 2019), for image classification tasks commonly used to evaluate domain adaptation methods. Our DAMNets yield a significant accuracy boost in the presence of large domain shifts, particularly when using more than one source domain. A more comprehensive evaluation on all datasets is provided in Appendix D.

| | Digits: MNIST (M), MNIST-M (MM), SVHN (S) | | | | | | Office-Home: Art (A), Clipart (C), Product (P), Real (R) | | | | | | | | |
|---|---|---|---|---|---|---|---|---|---|---|---|---|---|---|---|
| Source(s) | M | S | M | MM | M,MM | M,MM | A | C | C | R | A | C | P | C,P | A,C,P |
| Target | MM | M | S | S | S | S⋆ | P | P | A | A | R | R | R | R | R |
| No DA | 52.25 | 54.90 | 25.57 | 27.49 | 33.52 | 22.88 | 37.03 | 36.67 | 29.65 | 50.91 | 53.12 | 43.03 | 46.42 | 59.39 | 58.72 |
| DANN | 76.66 | 73.90 | 31.69 | 37.43 | 44.16 | 49.02 | 58.50 | 70.50 | 47.93 | 57.68 | 56.40 | 57.90 | 62.30 | 70.53 | 72.00 |
| RPT | 82.24 | 78.70 | 34.72 | 37.90 | *n/a* | *n/a* | 54.51 | 63.18 | 47.32 | 51.90 | 52.15 | 55.05 | 62.16 | *n/a* | *n/a* |
| **Ours** | **88.80** | **81.30** | **37.95** | **39.41** | **51.83** | **79.45** | **59.30** | **77.50** | **51.24** | **60.74** | **59.90** | **62.70** | **65.00** | **72.25** | **77.65** |
| *On TD* | *96.21* | *99.26* | *89.23* | *89.23* | *89.23* | *96.07* | *87.66* | *87.66* | *64.42* | *64.42* | *77.80* | *77.80* | *77.80* | *77.80* | *77.80* |

2010), which contains three domains: Amazon, DSLR, and Webcam; **Office-Home** (Venkateswara et al., 2017), with domains Art, Clipart, Product, and Real; and **VisDA17** (Peng et al., 2018), with Synthetic and Real images. As all these are well studied benchmark datasets, we provide full descriptions and image examples evidencing the different degrees of domain shift in Appendix B.

**Setup.** As discussed in Section 3, our method is general and can work with any feed-forward network architecture. To showcase this, for the digit recognition datasets, we apply it to the LeNet and SVHNet architectures (Ganin & Lempitsky, 2015), which are very simple convolutional networks, well suited for small images. Following Ganin & Lempitsky (2015), we employ LeNet when using the synthetic datasets MNIST and MNIST-M as source domains, and SVHNet when SVHN acts as source domain. We extend these architectures to multibranch ones by defining the computational units as the groups of consecutive convolution, pooling and non-linear operations defined in the original model. For simplicity, we use as many branches within each computational unit as we have domains, and all branches from a computational unit follow the same architecture, which we provide in Appendix A, Figures 1 and 2. As backbone network to process all the rest of the datasets, we use a ResNet-50 (He et al., 2016), with the bottleneck layer modification of Rozantsev et al. (2019). While many multibranch configurations can be designed for such a deep network, we choose to make our gated computational units coincide with the layer groupings defined in He et al. (2016), namely conv1, conv2_x, conv3_x, conv4_x, and conv5_x. The resulting multibranch network is depicted in Appendix A, Figure 4. We feed our DAMNets images resized to $224 \times 224$ pixels, as expected by ResNet-50.

**Results.** The results for the digit recognition and Office-Home datasets are provided in Table 1. Results for Office and VisDA17 datasets are presented in Appendix D. Our approach outperforms the baselines in *all* cases.

For the **Digits** datasets, in addition to the traditional two-domain setup, we also report results when using two source domains simultaneously. Note that the reference method RPT (Rozantsev et al., 2019) does not apply to this setting, since it was designed to transform a *single* set of source parameters to the target ones. Altogether, our method consistently outperforms the others. Note that the first two columns correspond to the combinations reported in the literature. We believe, however, that the SVHN ▷ MNIST one is quite artificial, since, in practice, one would typically annotate simpler, synthetic images and aim to use real ones at test time. We therefore also report synthetic ▷ SVHN cases, which are much more challenging. The multi-source version of our method achieves a significant boost over the baselines in this scenario. To further demonstrate the potential of our approach in this setting, we replaced its backbone with the much deeper ResNet-50 network and applied it on upscaled versions of the images. As shown in the column indicated by a ⋆, this allowed us to achieve an accuracy close to 80%, which is remarkable for such a difficult adaptation task.

On **Office-Home**, the gap between DAMNet and the baselines is again consistent across the different domain pairs. Note that, here, because of the relatively large number of classes, the overall performance is low for all methods. Importantly, our results show that we gain performance by training on more than one source domain, and by leveraging all synthetic domains to transfer to the real one, our approach reaches an accuracy virtually equal to that of using full supervision on the target domain. Despite our best efforts, we were unable to obtain convincing results for RPT using the authors' publicly available code, as results for this dataset were not originally reported for RPT.

**Gate dynamics.** To understand the way our networks learn the domain-specific branch assignments, we track the state of the gates for all computational units over all training epochs. In Figure 4,

we plot the corresponding evolution of the gate activations for the DSLR+Webcam ▷ Amazon task on Office. Note that our DAMNet leverages different branches over time for each domain before reaching a firm decision. Interestingly, we can see that, with the exception of the first unit, which performs low-level computations, DSLR and Webcam share all branches. By contrast, Amazon, which has a significantly different appearance, mostly uses its own branches, except in two computational units. This evidences that our network successfully understands when domains are similar and can thus use similar computations.

## 4.4 OBJECT DETECTION

We evaluate our method for the detection of drones from video frames, on the UAV-200 dataset (Rozantsev et al., 2018), which contains examples of drones both generated artificially and captured from real video footage. Full details and example images are provided in Appendix B.3

**Setup.** Our domain adaptation leverages both the synthetic examples of drones, as source domain, and the limited amount of annotated real drones, as target domain, as well as the background negative examples, to predict the class of patches from the validation set of real images. We follow closely the supervised setup and network architecture of Rozantsev et al. (2019), including the use of AdaDelta as optimizer, cross-entropy as loss function, and average precision as evaluation metric. Our multibranch computational units are defined as groupings of successive convolutions, nonlinearities, and pooling operations. The details of the architecture are provided in Appendix A, Figure 3.

| Method | Average precision |
|---|---|
| No adaptation | 0.377 |
| DANN (Ganin & Lempitsky, 2015) | 0.715 |
| ADDA (Tzeng et al., 2017) | 0.731 |
| Two-stream (Rozantsev et al., 2018) | 0.732 |
| RPT (Rozantsev et al., 2019) | 0.743 |
| DAMNet | **0.792** |

Table 2: Average precision of our DAMNet approach with several other reference methods, for domain adaptation from synthetic to real images of drones.

**Results.** Our method considerably surpasses all the others in terms of average precision, as shown in Table 2, thus validating DAMNets as effective models for leveraging synthetic data for domain adaptation in real-world problems.

## 4.5 DAMNET AS A GENERAL FEATURE EXTRACTOR

We validate the effectiveness of our method as a feature extractor, by combining it with the Maximum Classifier Discrepancy (MCD) method of Saito et al. (2018). As MCD operates on the extracted encodings, we replace the encoding strategy that MCD uses, which is the same as DANN, with our DAMNet. Or, in other words, we replace the domain classifier in our approach with the corresponding MCD term. Specifically, we use a single computational unit with two branches, each of which replicates the architectures proposed in Saito et al. (2018).

Table 3: We boost the method of Saito et al. (2018) by replacing their feature extraction with our DAMNets.

| | MCD accuracy | |
|---|---|---|
| | No DAMNet | with DAMNet |
| MNIST-M ▷ SVHN | 38.54 | 41.51 |
| DSLR ▷ Amazon | 67.24 | 67.81 |
| Webcam ▷ Amazon | 64.33 | 66.19 |
| Clipart ▷ Real | 63.50 | 63.87 |

We present the results of combining MCD with DAMNet in Table 3. In all tested scenarios, we obtain improvements over using MCD as originally proposed.

## 4.6 BRANCH ARCHITECTURES

To obtain more insights about specific branch decisions, we evaluate the effects of adding extra branches to the network, as well as using branches with different capacities.

### 4.6.1 BRANCHES WITH DIFFERENT CAPACITIES

When computational units are composed of branches of different capacities, DAMNets often assign branches with more capacity to more complex domains. To exemplify this, we trained a modified multibranch SVHNet for adaptation between MNIST and SVHN. Instead of the identical branches originally used, we replace the second branch in each computational unit with a similar branch where the convolution operation is performed by 1x1 rather than 5x5 kernels. These second branches, with

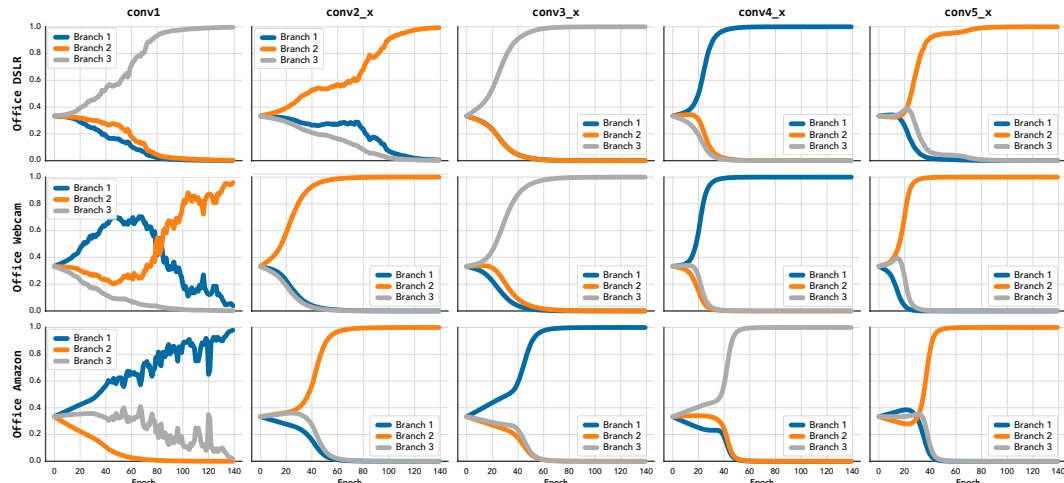

Figure 4: **Evolution of the gates' activations** for each of the computational units in a multibranch ResNet-50 network, for the Office DSLR + Webcam ▷ Amazon domain adaptation problem. In the top two rows, we show the gates for the source domains and in the bottom row for the target one. All branches are initialized to parameters obtained from a single ResNet-50 trained on ImageNet. Note how for the first computational unit, conv1, each domain chooses to process the data with different branches. In the remaining units, the two source domains, which have similar appearance, share all the computations. By contrast, the target domain still uses its own branches in conv3_x, and conv4_x to account for its significantly different appearance. When arriving at conv_5x, the data has been converted to a domain-agnostic representation, and hence the same branch can operate on all domains.

25 times fewer parameters each, are mostly used by the simpler domain—MNIST in this case. We provide the gate evolution that reflects this in Appendix C, Figures 5 and 6.

### 4.6.2 ADDITIONAL BRANCHES

We explore the effects of using more branches than domains, so as to provide the networks with alternative branches from where to choose. In particular, we explore the case where $K = D + 1$. We evaluate multibranch LeNet and ResNet architectures under this setting. We show the gate activation evolution in Appendix C, Figures 7 and 8. During the training process, we have observed that the networks quickly choose to ignore extra branches when $K > D$. This suggests that they did not contribute to the learning of our feature extraction. We did not find experimental evidence to support that $K > D$ is beneficial.

## 5 CONCLUSION

We have introduced a domain adaptation approach that allows for adaptive, separate computations for different domains. Our framework relies on computational units that aggregate the outputs of multiple parallel operations, and on a set of trainable domain-specific gates that adapt the aggregation process to each domain. Our experiments have demonstrated the benefits of this approach over the state-of-the-art weight untying strategy; the greater flexibility of our method translates into a consistently better accuracy.

Although we only experimented with using the same branch architectures within a computational unit, our framework generalizes to arbitrary branch architectures, the only constraint being that their outputs are of commensurate shapes. An interesting avenue for future research would therefore be to automatically determine the best operation to perform for each domain, for example by combining our approach with neural architecture search strategies.

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

# Appendices

## A    MULTIBRANCH ARCHITECTURES

Below, we provide the network architectures and detailed building blocks of our Domain Adaptive Multibranch Networks, for the single source domain case (D = 2). Each computational unit is enclosed by dotted lines. The input and output shapes for all layer groupings are provided.

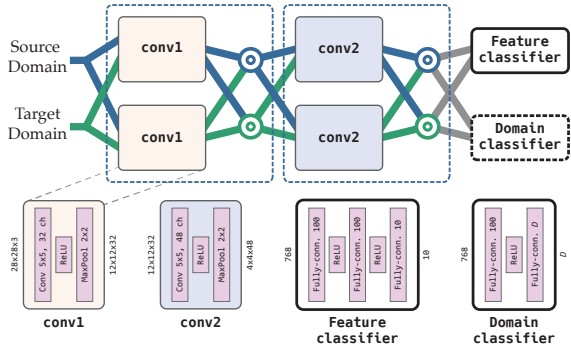

Figure 1: **Multibranch LeNet.** This architecture is a multibranch extension to the LeNet used by DANN (Ganin & Lempitsky, 2015).

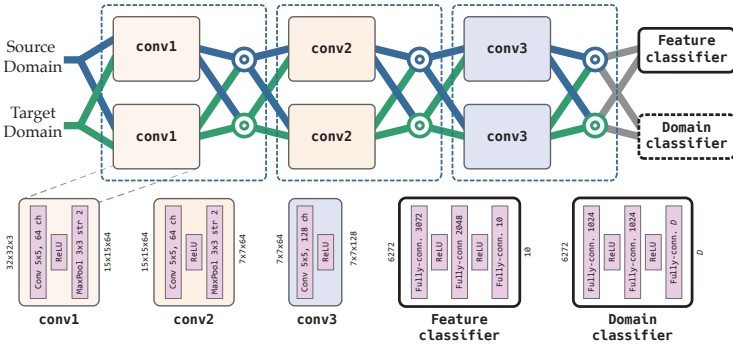

Figure 2: **Multibranch SVHNet.** This architecture is a multibranch extension to the SVHNet used by DANN (Ganin & Lempitsky, 2015).

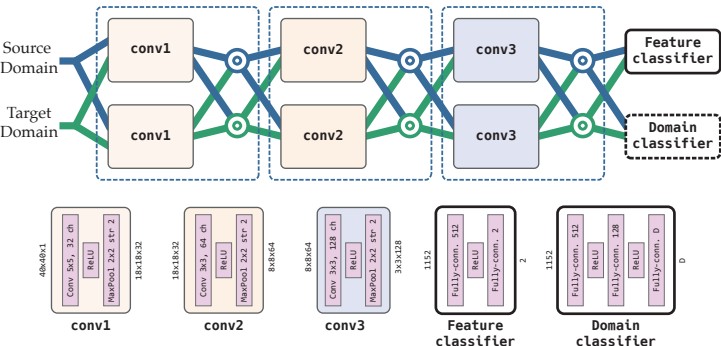

Figure 3: **Multibranch architecture for drone detection.** This architecture is a multibranch extension to the one used by Rozantsev et al. (2019).

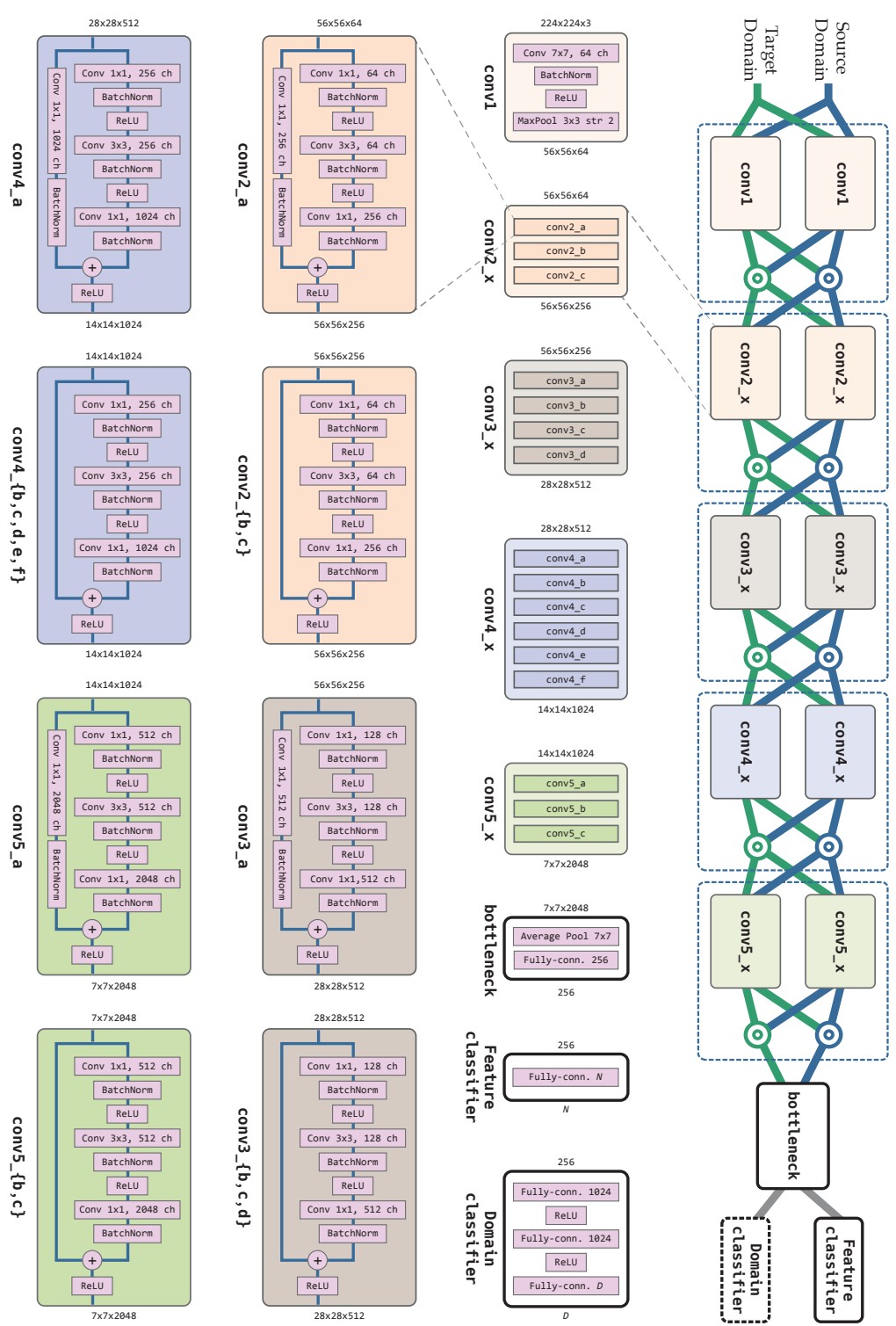

Figure 4: **Multibranch ResNet-50.** This architecture is adapted from the original ResNet-50 (He et al., 2016). We preserve the groupings described in the original paper (He et al., 2016). $N$ denotes the number of classes in the dataset.

# B BENCHMARK DATASET DESCRIPTIONS

## B.1 DIGIT RECOGNITION

**MNIST (LeCun et al., 1998)** consists of black and white images of handwritten digits from 0 to 9. All images are of size $28 \times 28$ pixels. The standard training and testing splits contain 60,000 and 10,000 examples, respectively.

**MNIST-M (Ganin & Lempitsky, 2015)** is synthetically generated by randomly replacing the foreground and background pixels of random MNIST samples with natural images. Its image size is $32 \times 32$, and the standard training and testing splits contain 59,001 and 9,001 images, respectively.

**SVHN (Netzer et al., 2011),** the Street View House Numbers dataset, consists of natural scene images of numbers acquired from Google Street View. Its images are also of size $32 \times 32$ pixels, and its preset training and testing splits are of 73,257 and 26,032 images, respectively. The SVHN images are centered at the desired digit, but contain clutter, visual artifacts, and distractors from its surroundings.

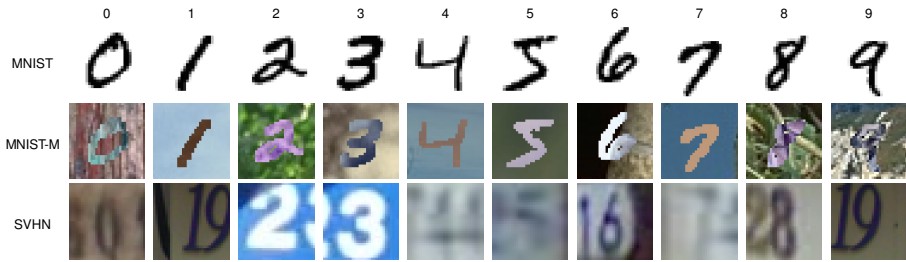

## B.2 OBJECT RECOGNITION

**Office (Saenko et al., 2010)** is a multiclass object recognition benchmark dataset, containing images of 31 categories of objects commonly found in office environments. It contains color images from three different domains: 2,817 images of products scraped from Amazon, 498 images acquired using a DSLR digital camera, and 795 images captured with a webcam. The images are of arbitrary sizes and aspect ratios.

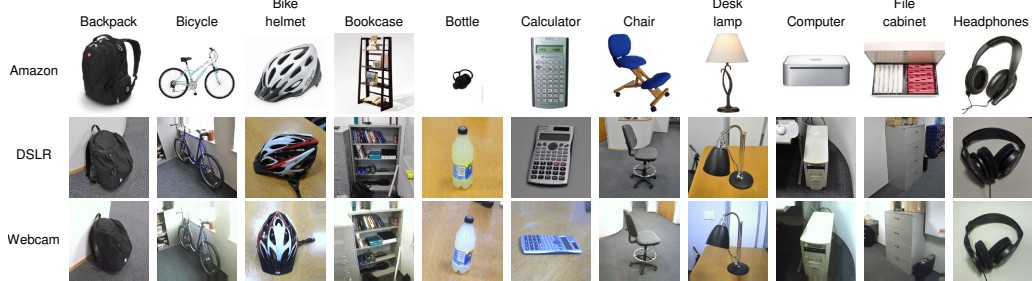

**Office-Home (Venkateswara et al., 2017)** comprises a larger corpus of color, arbitrarily-sized images from 65 different classes of objects commonly found in office and home environments, coming from four different domains. It contains 2,427 images extracted from paintings (Art), 4,365 clipart images (Clipart), 4,439 photographs of products (Product), and 4,357 pictures captured with a regular consumer camera (Real world).

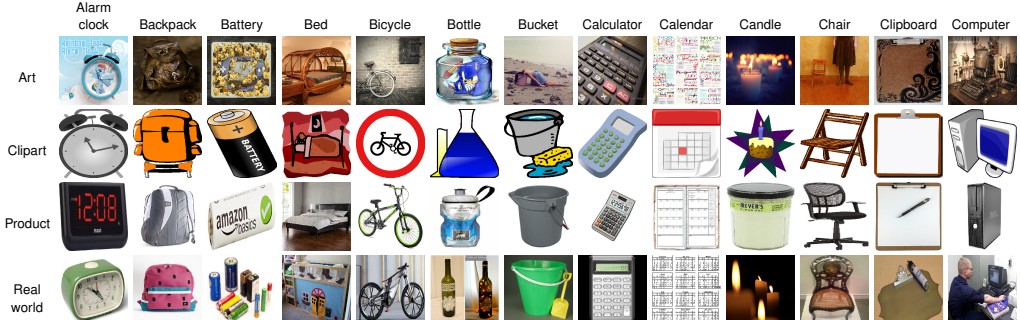

**VisDA 2017 (Peng et al., 2018)** includes images of diverse sizes from 12 different categories, coming from two different domains: 55,368 synthetic renders of 3D models, and 152,397 photographs of the real-world objects. It is larger than the other two datasets, and exhibits a more significant domain shift.

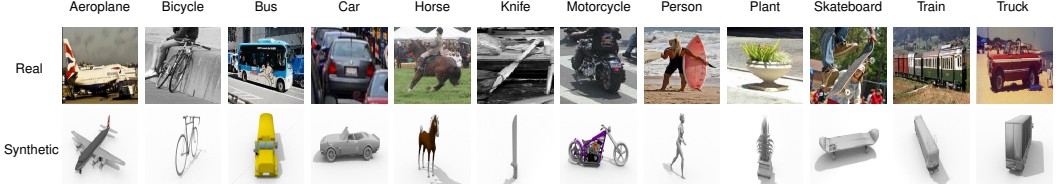

## B.3 OBJECT DETECTION

**UAV-200** aggregates 200 images of real drones and around 33,000 synthetic ones, as well as around 190,000 patches obtained from the background of the video, which do not contain drones, used as negative examples. All examples are of size $40 \times 40$ pixels. We evaluate performance on a validation set comprising 3,000 positive and 135,000 negative patches.

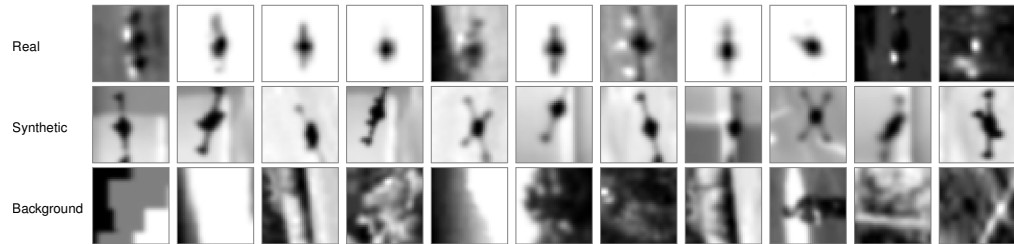

## C    ADDITIONAL EXPERIMENTS

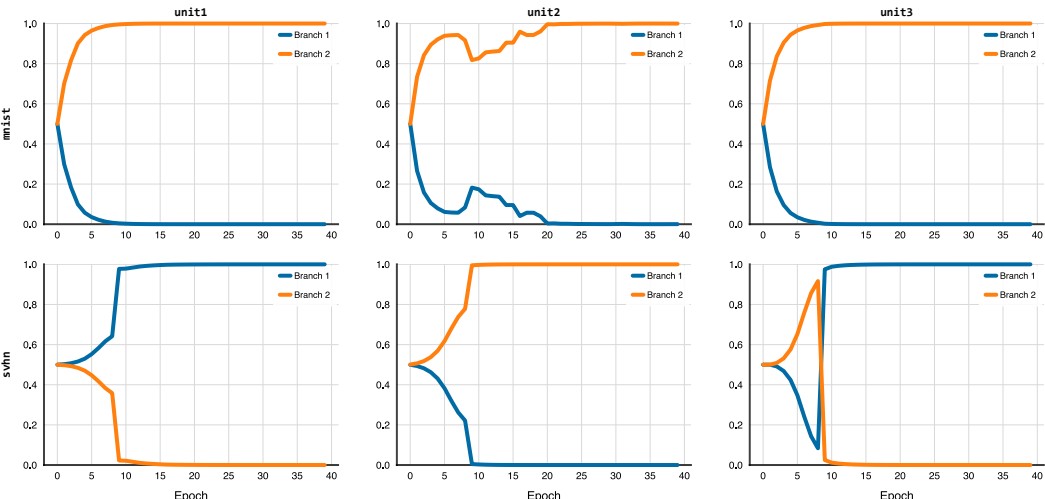

Figure 5: Gate evolution for a multibranch SVHN network with branches of different capacities. Branch 1 is the original branch that applies 5x5 convolutions to the image, whereas branch 2 is a similar architecture but with 1x1 convolutions instead. The network quickly recognizes that SVHN requires a more complex processing and hence assigns the respective branch to it for computational units 1 and 3.

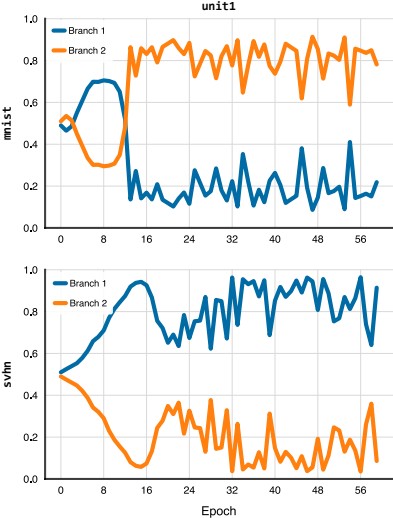

Figure 6: Gate evolution for a multibranch LeNet network with branches of different capacities. We have simplified the architecture to encapsulate the feature extraction into a single computational unit in this case. Similarly to the above, we modify the second branch for a simpler computation. The original branches apply convolution operations to extract 32 channels with a 5x5 kernel, and then to extract 48 channels from those with a 5x5 kernel. We replace them in the second branch with 24 channels 3x3 kernel and 48 channels 1x1 kernel convolutions, respectively, which yields commensurate shapes with the original branch, but with more than 20 times fewer parameters. Unlike in the above experiment, we do not force the gates to open or close. The network still assigns combinations of branches that reflect the difference in visual complexity of the domains.

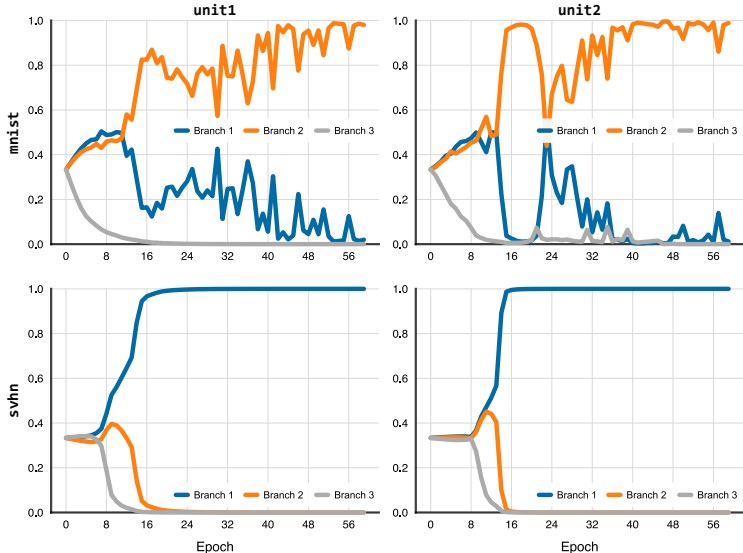

Figure 7: Effect of adding extra branches to a LeNet multibranch network. We augment the original multibranch LeNet with a third branch under the same branch architecture as the original one. The network rapidly decides to ignore this overparametrization. The additional branch does not have an effect in the final activation of the gates, nor does it help during training.

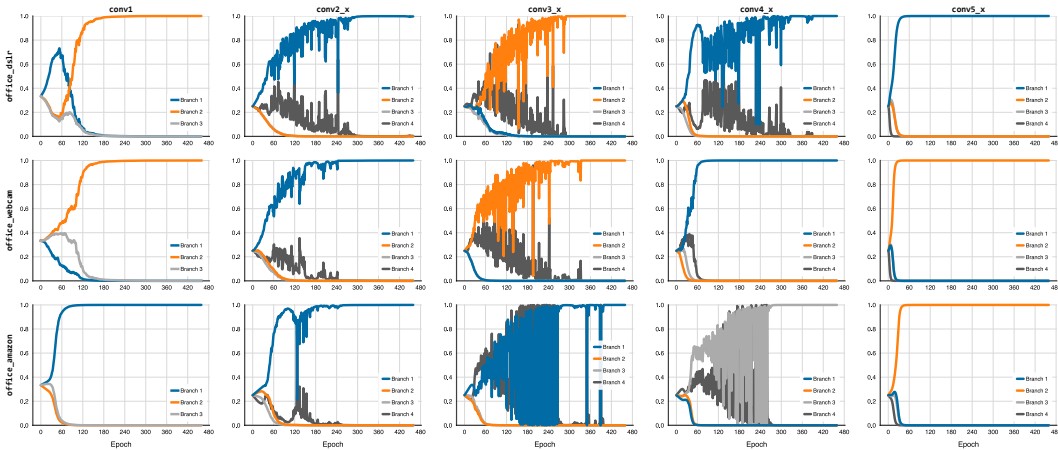

Figure 8: Augmenting a multibranch ResNet-50 has a similar effect as the above. One of the branches is discarded early on by each computational unit.

# D    FULL RESULTS

Table 4: **Domain Adaptation results.** We compare the accuracy of our DAMNet approach with that of DANN (Ganin & Lempitsky, 2015) and of RPT (Rozantsev et al., 2019), for image classification tasks commonly used to evaluate domain adaptation methods. As illustrated in Appendix B, different source and target domain combinations present various degrees of domain shift, and some combinations are clearly more challenging than others. Our DAMNets yield a significant accuracy boost in the presence of large domain shifts, particularly when using more than one source domain.

| Datasets | Source(s) ▷ Target | No DA | DANN | RPT | DAMNet | On TD |
|---|---|---|---|---|---|---|
| Digits | MNIST ▷ MNIST-M | 52.25 | 76.66[†] | 82.24 | **88.80** | *96.21* |
| | SVHN ▷ MNIST | 54.90 | 73.90[†] | 78.70[†] | **81.30** | *99.26* |
| | MNIST ▷ SVHN | 25.57 | 31.69 | 34.72 | **37.95** | *89.23* |
| | MNIST-M ▷ SVHN | 27.49 | 37.43 | 37.90 | **39.41** | *89.23* |
| | MNIST + MNIST-M ▷ SVHN | 33.52 | 44.16 | *n/a* | **51.83** | *89.23* |
| | MNIST + MNIST-M ▷ SVHN* | 22.88 | 49.02 | *n/a* | **79.45** | *96.07* |
| Office | Webcam ▷ DSLR | 93.60 | 99.20[†] | 99.40[†] | **99.62** | *95.20* |
| | Amazon ▷ DSLR | 32.80 | 79.10[†] | 82.70[†] | **84.14** | *95.20* |
| | DSLR ▷ Webcam | 90.45 | 97.70[†] | 98.00[†] | **98.11** | *98.49* |
| | Amazon ▷ Webcam | 34.67 | 78.90[†] | 81.50[†] | **85.28** | *98.49* |
| | Webcam ▷ Amazon | 41.42 | 62.80[†] | 63.60[†] | **65.67** | *85.11* |
| | DSLR ▷ Amazon | 34.47 | 63.60[†] | 64.70[†] | **64.82** | *85.11* |
| | DSLR + Webcam ▷ Amazon | 45.82 | 64.86 | *n/a* | **68.87** | *85.11* |
| Office-Home | Art ▷ Product | 37.03 | 58.50 | 54.51 | **59.30** | *87.66* |
| | Clipart ▷ Product | 36.67 | 70.50 | 63.18 | **77.50** | *87.66* |
| | Clipart ▷ Art | 29.65 | 47.93 | 47.32 | **51.24** | *64.42* |
| | Real world ▷ Art | 50.91 | 57.68 | 51.90 | **60.74** | *64.42* |
| | Art ▷ Real world | 53.12 | 56.40 | 52.15 | **59.90** | *77.80* |
| | Clipart ▷ Real world | 43.03 | 57.90 | 55.05 | **62.70** | *77.80* |
| | Product ▷ Real world | 46.42 | 62.30 | 62.16 | **65.00** | *77.80* |
| | Clipart + Product ▷ Real world | 53.39 | 70.53 | *n/a* | **72.25** | *77.80* |
| | Art + Clipart + Product ▷ Real world | 58.72 | 72.00 | *n/a* | **77.65** | *77.80* |
| VisDA 2017 | Synthetic ▷ Real | 35.46 | 59.90 | 61.10 | **61.40** | *84.72* |
| | Real ▷ Synthetic | 51.12 | 83.10 | 82.15 | **85.20** | *99.34* |
| UAV-200 | Synthetic ▷ Real* | 0.377 | 0.715 | 0.743 | **0.792** | *0.858* |

[†]Accuracy reported in Ganin & Lempitsky (2015) and Rozantsev et al. (2019)
*Evaluated with a ResNet-50
*Results reported as Average Precision

