# OpenReview forum: "Domain Adaptive Multibranch Networks"
_ICLR.cc/2020/Conference — Accept (Poster)_

### Official Review · AnonReviewer3 · 2019-10-09
**Official Blind Review #3**

**Rating:** 3

**Review:**

In this paper, the authors proposed to address the information asymmetry between domains in unsupervised domain adaptation. Innovatively, they resort to a multiflow network where each domain adaptatively selects its own pipeline. I quite appreciate the idea itself, while there are many essential issues to be addressed first.

Pros:
-	The way tackling the information asymmetry, or untie weights, between domains is novel and interesting.
-	The proposed network/framework can be easily extended to the multi-task setting, or multi-source/multi-target domain adaptation.
-	The paper is well-written and easy to follow.

Cons:
-	The most critical downside of this paper is its insufficient experiments to support the whole idea, where we will detail in the next.

Experimental issues:
-	Comparison with other state-of-the-art UDA methods (e.g., CDAN) is a must. This paper improves UDA in terms of adaptive parameters sharing, which is completely independent from most of the UDA contributions (including the DANN you compared) which improve the distribution alignment between feature representations. Therefore, it is imperative to compare that line of SOTA methods, otherwise why should we consider adaptative parameters sharing instead of distribution alignment? At best, the proposed multiflow network combined with the SOTA feature alignment method (e.g., CDAN other than DANN) should be considered and expected to beat CDAN itself.
-	Many ablation studies or hyperparameter sensitivity analyses are missing.
o	How do you determine the number of parallel flows, i.e., K? Is it possible that 3 or 4, more than 2 flows, are better even in the UDA between two domains?
o	Do you try any other possibilities of grouping a computational unit, and how will different configurations influence the performance?
o	Is there a possibility that none of the gates in the final layer is activated? Do you need some constraints?
-	Since the authors mentioned the potential of the multi-flow network in adaptation between multiple domains, it is necessary to investigate multi-source or multi-target domain adaptation. Only in this case may the significance of different K values be demonstrated.
-	The baseline results in Table 1 are not comparable to some reported papers, and even lower than those reported in other UDA papers.

**Experience Assessment:**

I have published in this field for several years.

**Review Assessment: Checking Correctness Of Derivations And Theory:**

I carefully checked the derivations and theory.

**Review Assessment: Checking Correctness Of Experiments:**

I carefully checked the experiments.

**Review Assessment: Thoroughness In Paper Reading:**

I read the paper thoroughly.

---

> ### Author Response · Authors · 2019-11-13
> **Comments incorporated in revised version**
>
> Thank you for your thorough review. Here are our responses:
>
> > Experimental issues:
> > -     Comparison with other state-of-the-art UDA methods (e.g., CDAN) is a must.
> > This paper improves UDA in terms of adaptive parameters sharing, which is
> > completely independent from most of the UDA contributions (including the DANN
> > you compared) which improve the distribution alignment between feature
> > representations. Therefore, it is imperative to compare that line of SOTA
> > methods, otherwise why should we consider adaptative parameters sharing instead
> > of distribution alignment? At best, the proposed multiflow network combined
> > with the SOTA feature alignment method (e.g., CDAN other than DANN) should be
> > considered and expected to beat CDAN itself.
>
> As acknowledged by R3, the contribution of our method happens at the representation extraction level, and is agnostic to the distribution alignment itself.
> Our comparison to RPT was motivated by the fact that it is the closest approach to ours, and since RPT relies on DANN, we also used it. However, our approach can indeed be employed with other distribution alignment strategies, and after the submission deadline, we have experimented with the Maximum Classifier Discrepancy (MCD) term of Saito et al., 2018. Our new results evidence that this can further improve our performance and consistently outperforms the original MCD.
>
> > -     Many ablation studies or hyperparameter sensitivity analyses are missing.
> > o     How do you determine the number of parallel flows, i.e., K? Is it possible
> > that 3 or 4, more than 2 flows, are better even in the UDA between two domains?
>
> We are including experiments studying this in the reviewed version of the paper. During the training process, we have observed that the networks quickly choose to ignore extra flows when K > D. This suggests that they did not contribute to the learning of our feature extraction. We did not find experimental evidence to support that K > D is beneficial.
>
> > o     Do you try any other possibilities of grouping a computational unit, and
> > how will different configurations influence the performance?
>
> There are arbitrarily many ways to group layers into computational units, and in our experiments, we used the blocks naturally emerging from the original architecture, such as the convolutional blocks defined in ResNet-50.
> An extensive evaluation would require much more time than that available for this rebuttal, and we believe that our current results already show the benefits of our approach.
>
> > o     Is there a possibility that none of the gates in the final layer is
> > activated? Do you need some constraints?
>
> It is theoretically possible that none of the gates be activated.  In practice, we have found that both the classification signal (for supervised domains) and the feature alignment (for unsupervised ones) are enough to prevent this from happening.
>
> > -     Since the authors mentioned the potential of the multi-flow network in
> > adaptation between multiple domains, it is necessary to investigate
> > multi-source or multi-target domain adaptation. Only in this case may the
> > significance of different K values be demonstrated.
>
> We have included experimental results using two and three source domains. The unsupervised multi-target domain problem is significantly more difficult and will require additional constraints. While we believe it to be an interesting problem, we consider it more suitable for future work.
>
> > -     The baseline results in Table 1 are not comparable to some reported papers,
> > and even lower than those reported in other UDA papers.
>
> We have either used the results available on the respective references, when available, or used publicly available code to generate results under the suggested experimental conditions (number of training epochs, batch size, data preprocessing, optimization algorithm, learning rate). We have been careful not to introduce any biases from our method so as to make meaningful comparisons.  It is, however, possible that under different experimental conditions other papers arrive to different results.
>
> We have updated the manuscript to reflect the above changes.
>
> Thank you for your input.

---

### Official Review · AnonReviewer2 · 2019-10-21
**Official Blind Review #2**

**Rating:** 6

**Review:**

After Discussion Period:

I stick to my original score. My issues are largely resolved.

----
The submission is using adaptive computation graphs for domain adaptation. Multi-flow network is the main architectural element proposed in the submission. And, it is composed of parallel blocks of computations which aggregated using weighted summation with learnable weights. The domain adaptation is performed by setting different weights for source and target dataset. The adaptive weights and network parameters are all learned jointly by minimizing the combination of classification loss and domain difference loss.

Although the idea of adaptive computation is not novel and has been explored, their application to the domain adaptation problem is novel to the best of my knowledge. Moreover, the proposed method is sensible and technically sound.

The submission talks about different amount of computation needed per domain as an intuition behind the method. This is sensible and intuitive; however, it has not been experimented. The paper uses the same amount of layers for all domains making the amount of computation exactly same. It would be interesting to see the performance when different paths actually lead different computations. For example, parallel blocks can have different number of layers etc.

The submission only provides result for RPT and DANN. These are clearly not state-of-the-art domain adaptation methods. Proposed method does not necessarily need to have state-of-the-art adaptation results to be accepted, but not reporting what state-of-the-art performance is makes the experimental results incomplete.

Figure 4 suggests that there is no real parameter sharing at the end of the training. And, all domains have different computations. Authors should try to explain this behaviour since it is quite counter-intuitive.

In summary, proposed method is somewhat novel, interesting and seems to be working well. Improved discussion on the experimental study is definitely needed.


**Experience Assessment:**

I have published in this field for several years.

**Review Assessment: Checking Correctness Of Derivations And Theory:**

I carefully checked the derivations and theory.

**Review Assessment: Checking Correctness Of Experiments:**

I assessed the sensibility of the experiments.

**Review Assessment: Thoroughness In Paper Reading:**

I read the paper thoroughly.

---

> ### Author Response · Authors · 2019-11-13
> **Comments incorporated in revised version**
>
> Thank you for your analysis. Here are our comments:
>
> > Although the idea of adaptive computation is not novel and has been explored,
> > their application to the domain adaptation problem is novel to the best of my
> > knowledge. Moreover, the proposed method is sensible and technically sound.
>
> > The submission talks about different amount of computation needed per domain
> > as an intuition behind the method. This is sensible and intuitive; however,
> > it has not been experimented. The paper uses the same amount of layers for
> > all domains making the amount of computation exactly same. It would be
> > interesting to see the performance when different paths actually lead
> > different computations. For example, parallel blocks can have different
> > number of layers etc.
>
> We have included results showing the behavior under this setting in the revised version of the paper. In particular, we study both using different capacities for the flows and incorporating additional flows to have more flows than domains. The results are provided in Appendix C and show that, in the first case, the more complex domains tend to use flows with higher capacity, and in the second case, that learning tends to discard some flows when there are more than domains.
>
> > The submission only provides result for RPT and DANN. These are clearly not
> > state-of-the-art domain adaptation methods. Proposed method does not
> > necessarily need to have state-of-the-art adaptation results to be accepted,
> > but not reporting what state-of-the-art performance is makes the experimental
> > results incomplete.
>
> Our comparison to RPT was motivated by the fact that it is the closest approach to ours. Since RPT relies on the same domain classifier as DANN, DANN came as a natural baseline. However, we will further report the state-of-the-art performance.
>
> Note that most of the SOTA techniques can also benefit from our approach. To evidence this, we performed additional experiments by replacing the DANN domain classifier in our approach with the Maximum Classifier Discrepancy (MCD) term of Saito et al., 2018. This further improves our results and our approach consistently outperforms the original MCD.
>
> > Figure 4 suggests that there is no real parameter sharing at the end of the
> > training. And, all domains have different computations. Authors should try to
> > explain this behaviour since it is quite counter-intuitive.
>
> We believe that there was some confusion. In Fig. 4, each row indicates how much each domain uses each flow. In each column, the same color indicates the same flow. As such, at the end of the training, Fig. 4 shows that, for example, all domains share the same flows in computational units conv2_x and conv5_x. We acknowledge that, in conv1, each domain uses its own private flow. This, we believe, confirms our intuition: Initially, the domains need to undergo different computations, because of their appearance differences, but can then share some of the following computations in later stages of the network. We hope that this clarifies the reviewer's concern.
>
> > In summary, proposed method is somewhat novel, interesting and seems to be
> > working well. Improved discussion on the experimental study is definitely
> > needed.
>
> We have updated the manuscript to reflect the above changes.
>
> Thank you for your input.

---

### Official Review · AnonReviewer1 · 2019-10-23
**Official Blind Review #1**

**Rating:** 8

**Review:**

The paper provides an unsupervised domain adaptation approach
in the context of deep learning. The motivation is clear, related work
sufficient and experimental settings and results convincing.
I have only very minor comments:
- I would prefer to get the paper additionally linked to a few more
  transfer learning techniques out of the deep learning domain
  which is important as well
- do you really need to call it (multi) flow network .... - a flow network
  is a well established concept in algorithmics and refers to a graph problem
  ... to avoid name clashes ...
- in the references you have provided back links to the pages where the references
  are used - this is handy but also confusing and a bit unusual - I think it was not part
  of the standard template
- please avoid using arxiv references but replace them by reviewed material. In parts
  I am willing to accept such kind of gray literature provided by well known authors but
  this should not become a standard habit
- I am happy to see that the code will be published - I hope this is really done, because
  from the material it maybe hard to reconstruct the method

**Experience Assessment:**

I have published one or two papers in this area.

**Review Assessment: Checking Correctness Of Derivations And Theory:**

I assessed the sensibility of the derivations and theory.

**Review Assessment: Checking Correctness Of Experiments:**

I assessed the sensibility of the experiments.

**Review Assessment: Thoroughness In Paper Reading:**

I read the paper at least twice and used my best judgement in assessing the paper.

---

> ### Author Response · Authors · 2019-11-13
> **Comments incorporated in revised version**
>
> Thank you for your insights. Here are our responses:
>
> >- I would prefer to get the paper additionally linked to a few more transfer
> >learning techniques out of the deep learning domain which is important as well
>
> We focused our literature review on deep learning because our work proposes an
> approach for deep architectures. For the sake of completeness, we will
> nonetheless include the following papers:
>
> - Boosting for Transfer Learning [Dai07]
> - Covariate Shift by Kernel Mean Matching [Gretton09]
> - Domain adaptation via transfer component analysis [Pan10]
> - Unsupervised Visual Domain Adaptation Using Subspace Alignment [Fernando13]
> - Deep CORAL: Correlation Alignment for Deep Domain Adaptation [Sun16]
> - A DIRT-T approach to unsupervised domain adaptation [Shu18]
>
> >- do you really need to call it (multi) flow network .... - a flow network is
> >a well established concept in algorithmics and refers to a graph problem ...
> >to avoid name clashes ...
>
> We propose to rename our approach as Domain-Adaptive Multibranch Networks.
>
> >- in the references you have provided back links to the pages where the
> >references are used - this is handy but also confusing and a bit unusual - I
> >think it was not part of the standard template
>
> We will abide by the template and remove the back links.
>
> >- please avoid using arxiv references but replace them by reviewed material.
> >In parts I am willing to accept such kind of gray literature provided by well
> >known authors but this should not become a standard habit
>
> We have replaced the arxiv references with reviewed ones when available.
>
> >- I am happy to see that the code will be published - I hope this is really
> >done, because from the material it maybe hard to reconstruct the method
>
> Our code is currently stored in a private github repository, which will be set as public once the blind review period ends.
>
> We have updated the manuscript to reflect the above changes.
>
> Thank you for your input.

---

### Public Comment · ~Sumukh_Aithal_K1 · 2020-07-05
**Queries regarding the paper**

1. In Section 3.1 (page 4) of the paper it's mentioned that the plasticity is initially set to a small value and then increased as the training continues.But in the Section 4.2(Implementation Details), it is mentioned that the plasticity is decayed linearly as training progresses and plasticity is defined as 1-p.
Please clarify the above points.
2. Are all the branches initialized with same values? If yes then won't both the branches have the same parameters and behave like a single branch? If no, then how are they initialized?
And could you please let us know when you plan on releasing the code?

---

> ### Author Response · Authors · 2020-07-07
> **Plasticity term and Initialization details**
>
> Thanks for your interest in our work, and for giving us the chance to clarify on those issues.
>
> 1. You are correct, there is a mismatch in the usage of the term "plasticity". As you point out, in Section 3.1 we call the plasticity p and in in Section 4.2 plasticity is 1 - p. What we wanted to achieve with the plasticity term is to modulate the learning rate of the gates, so a larger value of the plasticity should semantically mean 'more flexible gates'. So, a "plasticity term that decays linearly as training progresses" means that the gates become less and less flexible over time. Maybe a less unfortunate wording would then be "the *parameter p* that controls the plasticity is initially set to a small value", which is then consistent with "plasticity (1-p) is decayed linearly as training progresses". So, as p increases, 1-p decreases.
>
> 2. For some experiments we initialize specific branches with parameters from other well-known networks such as ImageNet. We add a small random noise to avoid having the exact same parameters in different branches. Notice also that the gate parameters are part of each particular branch too, so differences in the gates will influence the training as well.

---

### Decision · Program_Chairs · 2019-12-19

**Decision:**

Accept (Poster)

**Comment:**

Although some criticism remains for experiments, I suggest to accept this paper.